# A Novel Approach for the Treatment of T Cell Malignancies: Targeting T Cell Receptor Vβ Families

**DOI:** 10.3390/vaccines8040631

**Published:** 2020-10-31

**Authors:** Jie Wang, Katarzyna Urbanska, Prannda Sharma, Reza Nejati, Lauren Shaw, Megan S. Lim, Stephen J. Schuster, Daniel J. Powell Jr.

**Affiliations:** 1Division of Hematologic Malignancies and Cellular Therapies, Duke Cancer Institute, Durham, NC 27710, USA; jie.wang416@duke.edu; 2Bristol-Meyers Squibb, 345 Park Ave, New York, NY 10154, USA; urbanskakasia@yahoo.com; 3Department of Pathology and Laboratory Medicine, Perelman School of Medicine, University of Pennsylvania, Philadelphia, PA 19104, USA; pranndas@pennmedicine.upenn.edu (P.S.); lashaw@pennmedicine.upenn.edu (L.S.); Megan.Lim@pennmedicine.upenn.edu (M.S.L.); 4Department of Pathology, Fox Chase Cancer Center, 333 Cottman Ave, Philadelphia, PA 19111, USA; reza.nejati@fccc.edu; 5Division of Hematology/Oncology, Department of Medicine, Perelman School of Medicine, University of Pennsylvania, Philadelphia, PA 19104, USA; stephen.schuster@uphs.upenn.edu

**Keywords:** T cell lymphoma, T cell receptor, antibodies, immunotherapy, CD64, immune receptor

## Abstract

Peripheral T cell lymphomas (PTCLs) are generally chemotherapy resistant and have a poor prognosis. The lack of targeted immunotherapeutic approaches for T cell malignancies results in part from potential risks associated with targeting broadly expressed T cell markers, namely T cell depletion and clinically significant immune compromise. The knowledge that the T cell receptor (TCR) β chain in human α/β TCRs are grouped into Vβ families that can each be targeted by a monoclonal antibody can therefore be exploited for therapeutic purposes. Here, we develop a flexible approach for targeting TCR Vβ families by engineering T cells to express a chimeric CD64 protein that acts as a high affinity immune receptor (IR). We found that CD64 IR-modified T cells can be redirected with precision to T cell targets expressing selected Vβ families by combining CD64 IR-modified T cells with a monoclonal antibody directed toward a specific TCR Vβ family in vitro and in vivo. These findings provide proof of concept that TCR Vβ-family-specific T cell lysis can be achieved using this novel combination cell–antibody platform and illuminates a path toward high precision targeting of T cell malignancies without substantial immune compromise.

## 1. Introduction

The T cell malignancies encompass a highly heterogeneous group of diseases and are generally associated with a poor prognosis. Combination chemotherapy approaches result in consistently poorer outcomes for T cell malignancies compared to B cell malignancies [1,2]. Non-targeted therapies for relapsed refractory T cell lymphomas including pralatrexate, romidepsin, and belinostat only achieve an overall response rate (ORR) of 25–27% [3]. Brentuximab Vedotin, an antibody drug conjugate to CD30, has an ORR of 86% in relapsed and refractory anaplastic large cell lymphoma (ALCL) but is not applicable to most other peripheral T cell lymphomas (PTCL) subtypes, which do not express CD30 [4]. Therefore, there is an urgent clinical need to develop novel alternative approaches to treat aggressive T cell malignancies.

The monoclonal antibody approach has been successful in targeting B cell malignancies. For example, rituximab has significantly improved the long-term overall survival in diffuse large B cell lymphoma, and CD20 directed monoclonal antibodies are often incorporated in immunochemotherapy regimens for B cell malignancies [5,6,7]. In contrast to the B cell malignancies, there is no universal antigen that can be safely targeted in the treatment of T cell malignancies. While hypogammaglobulinemia and recurrent infections may result from CD20-directed monoclonal antibody treatment, this complication can be overcome by giving exogenous intravenous immunoglobulin. Recently developed chimeric antigen receptor (CAR) modified T cells that show effective target T cell cytolysis in the laboratory target pan-T cell antigens [8,9,10]. However, the risks of wholesale T cell depletion resulting in severe immune compromise are not easily remediable, making these approaches very risky to translate to clinical use.

A different approach is to target the T cell receptor (TCR) itself. Since the TCR β-chain uses one of two constant regions, CAR T cells targeting one of the domains can effectively deplete only a subset of T cells [11]. While it is an improvement compared to targeting a pan T cell marker, the targeting of one of two TCR constant domains can still result in the depletion of approximately half of the T cell population and a substantial decrease in the T cell repertoire in the treated patient. Alternatively, the unique complementarity-determining region 3 on TCRs on lymphoma cells may be targeted by CAR T cells to provide high precision targeting; however, this approach is complicated by the necessity for sophisticated epitope screening and scFv panning, as well as the need for GMP production of a unique CAR lot for each individual patient, both of which are lengthy and costly [12]. Therefore, we reasoned that targeting the native TCR β-chain variable region by engineered T cells may be a more feasible clinical approach to maintain a relatively high degree of specificity for the malignant T cell clone with reduced cytolysis of healthy bystander T cells.

To effectively target the TCR variable region, we recognized that α/β TCRs can be subdivided into 24 functional Vβ families as determined by the sequence of the Vβ gene, and monoclonal antibodies may be used to differentially target Vβ families (Figure 1A,B). Therefore, we engineered normal donor T cells to express CD64, a high affinity Fc gamma receptor, and sought to redirect the engineered T cells by coupling them with monoclonal antibodies. We hypothesized that by using Vβ-family-specific monoclonal antibodies to redirect non-malignant T cells toward specific Vβ-family targets, we could target T cell malignancies, while avoiding pan-T cell depletion.

## 2. Materials and Methods

### 2.1. CD64 Immune Receptor Construction (CD64 IR)

Human CD64 DNA sequence was amplified from primary human monocytes using primers. After amplification and the insertion of 3′-Bam-H1 and 5′-Nhe-1 restriction sites, the PCR product was digested with Bam-HI and NheI enzymes and ligated into pELNS, a third-generation self-inactivating lentiviral expression vector, containing human CD28-CD3z signaling endodomains, under an EF-1a promoter. The resulting construct was designated pELNS CD64-IR-28z.

### 2.2. Recombinant Lentivirus Production

High-titer replication-defective lentiviral vectors were produced and concentrated, as previously described [13,14]. Briefly, 293T cells were co-transfected with VSV glycoprotein expression plasmid (pVSV-G), Rev expression plasmid (pRSV.REV), Gag/Pol expression plasmid (pMDLg/p.RRE), and pELNS transfer plasmid using Lipofectamine 2000 (Invitrogen, Carlsbad, CA, USA). The viral supernatant was harvested at 24 and 48 h post-transfection, sterile filtered using a 0.45 μM filter, and concentrated by ultracentrifugation at 26,000 rpm for 2 h at 4 °C. Aliquots of high titer lentivirus were stored at −80 °C until use.

### 2.3. Generation of Vβ12 TCR Transduced SupT1 Cell Line (SupT1-Vβ12)

Generation of the PG13 MART 1 DMF4 TCR packaging cell line was performed as previously described [15]. Retroviral supernatant was collected 24 h after seeding the PG-13 producer clone at 70% of confluence. Non-tissue culture treated 6-well plates were coated with retronectin and blocked with 2% BSA per manufacturer’s protocol (Takara, Shiga, Japan, Cat # T100A/B). An amount of 2.3 mL retroviral supernatant was added per well and centrifuged at 2000× *g* for 2 h, and supernatant aspirated. Target cell line SupT1 was plated at a density of 4 × 10^5^ cells/mL and centrifuged for 10 min at 1000× *g*. Retronectin retroviral transduction was repeated the following day, and the transduced cell line was stained with MART1 tetramer and evaluated by flow cytometry. Vβ family usage was verified using Vβ12-FITC antibody (clone S511, Thermo Scientific, Waltham, MA, USA, Cat # TCR2654). This antibody recognizes Vβ10 using the International Inmunogenetics (IMGT) nomenclature [16]. The transduced cells were enriched using MACS magnetic bead cell separation (Miltenyi Biotec, Bergisch Gladbach, Germany).

### 2.4. Primary T Cell Activation, Transduction, and Expansion

Primary human total T cells were isolated from healthy volunteer donors, following leukapheresis by negative selection, and purchased from the Human Immunology Core at University of Pennsylvania. All specimens were collected under a University Institutional Review Board-approved protocol, and written informed consent was obtained from each donor. T cells were cultured in complete media (RPMI 1640 supplemented with 10% heat inactivated low IgG fetal bovine serum (FBS), 100 U/mL penicillin, 100 ug/mL streptomycin sulfate) and stimulated with anti-CD3 and anti-CD28 monoclonal antibody coated beads (Invitrogen) on day 0 as described. Twenty-four hours after activation, T cells were transduced with lentiviral vectors at MOI of ~5. Human recombinant interleukin-2 (IL-2; Novartis) was added every other day to 100 IU/mL final concentration and a 0.5–1 × 10^6^ cells/mL cell density was maintained. Starting at day 11–12 of culture, no more IL-2 was added to allow the T cells to rest. Rested T cells were used in subsequent co-culture assays. 

### 2.5. Cytotoxicity Assays

51Cr release assays were performed as described. Target cells were labeled with 100 mCi 51Cr at 37 °C and TCR Vβ-family-specific monoclonal antibody (1 μg per million target cells) for 1.5 h in low IgG R10 media. Target cells were washed three times in PBS, resuspended in phenol red free RPMI with 5% low IgG FBS (CM) at 10^5^ cells/mL and 100 μL added per well of a 96-well U-bottom plate. Effector cells were washed twice in CM and added to wells at the given ratios. Plates were centrifuged to settle cells and incubated at 37 °C in a 5% CO2 incubator for 4 h. The supernatants were harvested, transferred to a luminex-plate (Packard), and counted using a 1450 Microbeta Liquid Scintillation Counter (Perkin-Elmer, Waltham, MA, USA). Spontaneous 51Cr release was evaluated in target cells incubated with medium alone. Maximal 51Cr release was measured in target cells incubated with SDS at a final concentration of 5% (v/v). Percent-specific lysis was calculated as (experimental—spontaneous lysis/maximal—spontaneous lysis) × 100.

### 2.6. Vβ-Family-Specific Antibodies and Specificity Determination

The TCR Vβ-family-specific monoclonal antibody reagents used are Vβ12-FITC (clone S511, Thermo Scientific Cat # TCR2654) and Vβ8(a)-FITC (Thermo Scientific Cat # TCR2648). The antibodies bind the SupT1-Vβ12 cell line and Jurkat cell line, respectively. Fluorescence-activated cell sorting (FACS) was performed on SupT1-Vβ12 and Jurkats after staining with Vβ12-FITC and Vβ8-FITC antibodies from Thermo Scientific, and TCR sequencing was performed on the sorted cells. Vβ12-FITC clone S511 antibody is verified to be specific for Vβ10, using the International Immunogenetics (IMGT) nomenclature. Vβ8(a)-FITC is specific for IMGT Vβ12 family [16]. In this paper, all TCR Vβ families are referred to by the name that the monoclonal antibody reagents are labeled, which corresponds to a different classification system. 

### 2.7. In Vivo Studies

All animals were purchased from the Stem Cell and Xenograft Core at the University of Pennsylvania. Six-to-12-week-old severe combined immunodeficient NOD/SCID)/γ-chain-/-(NSG) mice were bred, treated, and maintained under pathogen-free conditions in-house under University of Pennsylvania IACUC approved protocols. For the Winn assays, 6-to-12-week-old female NSG mice were inoculated subcutaneously with 1–2 million firefly luciferase (fLuc) positive Jurkat or SupT1-Vβ12 T cells in conjunction with the appropriate targeting Vβ antibody (1 μg per million malignant T cells), and with either untransduced primary human total T cells or T cells transduced with the CD64 IR construct. The primary T cell to target malignant T cell ratio was 1:1. Transduction efficiency ranged from 40–65%. After intraperitoneal injection of luciferin, bioluminescence imaging of mice was performed weekly for up to 35 days. Specifically, in a pilot Winn assay, Jurkats were used as the malignant T cell line, and 3 groups were compared with 2 mice per group. In the confirmatory Winn assay, both Jurkats and SupT1-Vβ12 were used as malignant T cell lines, and 3 groups of 5 mice each were compared.

### 2.8. Subjects and TISSUES

The University of Pennsylvania Hospital Pathology database was searched for cases indexed as a T cell malignancy from the period 2000–2017. After excluding all outside cases and bone marrow only cases, 140 tissue blocks and accompanying slides were retrieved and reviewed. Cases with controversial diagnoses or inadequate biopsy material were excluded from the study. The remaining 34 formalin fixed paraffin embedded (FFPE) blocks were cut into 5 × 5 micron scrolls for DNA extraction, and TCR β next generation sequencing (TRB NGS) was performed. Basic clinical diagnostic information regarding the patient cases were obtained from chart review. 

Of the 34 available tissue blocks, 12 samples demonstrated clonal dominance by TRB NGS (defined below), and 8 samples showed cell surface expression of α/β TCR by βF1 immunohistochemistry. TCR Vβ family usage was determined through raw sequence analysis. This retrospective study was conducted with institutional review board approval and in accordance with the Declaration of Helsinki. 

### 2.9. TCR Library Sequencing

TCR Vβ sequencing was performed by the Human Immunology Core Facility at the University of Pennsylvania. DNA was extracted from FFPE T cell malignancy patient samples or sorted cell lines using Qiagen Gentra Puregene cell kit and following manufacturer directions (Qiagen, Valencia, CA, USA, Cat. No. 158388). TCR Vβ-family-specific PCR was performed on all samples. The libraries for sequencing of the Illumina MiSeq platform were prepared using a cocktail of 23 Vβ families from framework region (FR)2 forward primers, and 13 Jβ region reverse primers modified from the BIOMED2 primer series [17]. Library quality was evaluated using Bioanalyzer 2100 (Agilent Technologies, Santa Clara, CA, USA) and quantified by Qubit Fluorometric Quantitation (Thermo Fisher Scientific, Grand Island, NY, USA). A sharp single band from Bioanalyzer analysis indicated a good quality library and was used for sequencing. The reading from Qubit using dsDNA HS (high sensitivity) assay kit (Cat. No. Q32851) was used to calculate the molarity of the library. Libraries were then loaded onto an Illumina MiSeq in the Human Immunology Core Facility at the University of Pennsylvania. Paired end kits of 2 × 300 bp were used for all experiments (Illumina MiSeq Reagent Kit v3, 600 cycle, Illumina Inc., San Diego, CA, USA, Cat. No. MS-102-3003). 

### 2.10. Raw Sequence Analysis

Raw sequence data (fastq files) were filtered as previously described [18]. Filtered sequence alignment and clone assembly were processed by MiXCR (version 2.1) and MiXCR (version 1.1.5). TCR clones were identified by their CDR3 nucleotide sequence. Dominant clones were defined as those that represent at least 3% of the T cell repertoire, and which are at least three times as frequent as the next most frequent clone.

## 3. Results

### 3.1. Constructing a Flexible Platform for TCR Vβ Family Targeting

In order to redirect T cells to antibody targets, we engineered T cells to express the high affinity Fc gamma receptor CD64, which in turn, binds to the Fc domain of a monoclonal antibody. We reasoned that the bound monoclonal antibody should redirect the engineered T cell to its target (Figure 1C). Human CD64 DNA sequence was amplified from primary human monocytes and inserted into pELNS, a third-generation self-inactivating lentiviral expression vector, containing human CD28 costimulatory and CD3z signaling endodomains (Figure 1D). The resulting CD64 immune receptor (IR) construct was transduced into normal donor T cells achieving 50–80% expression (Figure 1E).

After generating the CD64 IR-modified T cells, we next ascertained if different murine IgG subclasses can load onto the CD64 IR. Using flow cytometry, we observed successful antibody loading with murine IgG2a and IgG2b subclasses, but not murine IgG1 (Figure 1F). Of these loading antibodies, one was specific for TCR Vβ8, and another was specific for Vβ12. The specificity of these antibodies for the respective Vβ family was validated by using two cell lines: the Jurkat T cell leukemia cell line that naturally expresses the Vβ8 family TCR, and an engineered Vβ12 TCR expressing T cell line, made from SupT1 T cell lymphoblasts, which do not express a native functioning TCR, here, stably transduced to express a Vβ12+ MART-1-specific TCR (Figure 1G) [19].

### 3.2. CD64 IR-Modified T Cells Display Specific Cytolytic Function against TCR Vβ Families

We redirected CD64 IR-transduced T cells to target autologous T cells bearing specific TCR Vβ families using Vβ-family-directed monoclonal antibodies. We performed co-culture assays using CD64 IR T cells and the same number of autologous untransduced T cells. The target T cells were either “pre-painted” with Vβ-specific monoclonal antibody, or the CD64 IR-modified T cells were “pre-armed” with antibody. At 24 h, Vβ-family-specific depletion was assessed by flow cytometry. Both “pre-arming” of the effectors and “pre-targeting” of the targets resulted in Vβ-family-specific depletion when assayed by flow cytometry (Figure 2A,B). Since the presence of targeting antibody may have interfered with binding of the flow antibody for analysis, exaggerating the perceived result of target cell cytolysis, we performed chromium release assays to directly determine the specific cytolysis of α/β TCR expressing T cell lines, which effectively eliminates the effect of “epitope masking”.

Using the SupT1-Vβ12 cell line that was engineered for TCR Vβ12 expression (Figure 1G), we performed four-hour chromium release assays to determine Vβ family target specificity. At effector to target ratios ≥ 5:1, specific cytolysis of SupT1-Vβ12 cells was achieved only when a Vβ12-specific monoclonal antibody was applied (Figure 2C). Significantly less cytolysis was seen when a control antibody of the same isotype—but directed against Vβ8—was used.

Next, we tested whether CD64 IR T cells coupled with a specific Vβ family antibody could lyse a malignant T cell line expressing its natural α/β TCR. For this, we used the Jurkat T cell leukemia cell line that expresses the Vβ8 family TCR. While known to have an intact TCR and T cell signaling system, Jurkats are derived from a T cell leukemia patient in relapse, and also harbor several mutations in TP53, making them a good model for an aggressive T cell malignancy [20,21]. In four-hour chromium release assays using Vβ8+ Jurkat cells as targets, we again observed Vβ-family-specific cytolysis. A significantly greater percent specific cytolysis was seen when Vβ8 antibody was applied, compared to when the Vβ12 control antibody was used, or when no antibody was used (Figure 2C).

### 3.3. CD64 IR T Cells Lyse Target T Cells by CD64 IR TCR Activation

To test the mechanism of targeted T cell cytolysis, we created a signaling deficient CD64-dz IR construct, which has the same extracellular structure as the CD64 IR construct but lacks the intracellular CD3ζ T cell signaling domain (Figure 1D). We determined the transduction efficiency of the CD64-dz IR construct and confirmed successful Vβ antibody loading (Figure 1E,F). If CD64 IR T cells induced target T cell cytolysis by CD64-IR T cell activation, then target T cell cytolysis would not be seen if T cell signaling in the CD64 IR T cells is abrogated. On the other hand, if target T cell cytolysis occurred due to target T cell activation-induced cell death through antibody engagement of the target T cell TCR, then abrogation of T cell signaling in the effector arm would not abrogate the cytolytic effect seen. When CD64-dz T cells were cultured with SupT1-Vβ12 or Jurkat target T cells, only background target T cell cytolysis was observed, even in the presence of Vβ-specific antibodies (Figure 2C). This suggests that the mechanism of target T cell lysis of these T cell lines is through CD64-IR T cell engagement of the target TCR Vβ family through the specific monoclonal antibody, which initiates effector T cell activation through the CD3ζ T cell signaling domain.

### 3.4. Vβ Antibody in Conjunction with CD64 IR T Cells Prevents T Cell Malignancy Outgrowth In Vivo

The antitumor efficacy of CD64 IR T cells with Vβ-specific antibody was evaluated in vivo. In a pilot Winn assay designed to help establish experimental design, immunodeficient NSG mice were co-injected subcutaneously with firefly luciferase-transfected Jurkat cells that naturally express TCR Vβ8 and CD64 IR T cells with or without addition of Vβ8 antibody. Control mice were injected with the Jurkat cell line only. Co-administration of CD64 IR T cells slowed the growth of the Jurkat cells compared to that seen in the control mice; however, addition of Vβ8 antibody was seen to augment the response to CD64 IR T cells and significantly inhibit tumor growth compared to the Jurkat cell only control group (Figure 3A, *p* = 0.0079).

Based on results of this pilot study, an in vivo experiment was performed to ascertain if Vβ-family-specific targeting antibody must work in conjunction with CD64 IR T cells, rather than untransduced T cells, to produce a significant anti-tumor effect. Immunodeficient mice were injected with firefly luciferase-transfected Jurkat cells alone or with the targeting Vβ8 antibody combined with CD64 IR T cells or control untransduced T cells from the same donor. Tumor outgrowth was measured by bioluminescence and compared to the untreated control group. Vβ8 antibody administration in conjunction with CD64 IR T cell transfer was required to prevent tumor Jurkat cancer cell outgrowth; Vβ8 antibody in conjunction with untransduced T cells had no statistically significant impact (Figure 3B, *p* = 0.13), demonstrating that, in the absence of CD64 IR T cells, the Vβ8 antibody was ineffective. We next tested this treatment in a different model using SupT-1 cells that were engineered to express a clinically tested Vβ12 family TCR (referred to as SupT1-Vβ12) as target cells. Vβ12 antibody in conjunction with CD64 IR T cells controlled the outgrowth of SupT1-Vβ12 in vivo, while Vβ12 antibody in conjunction with untransduced T cells did not (Figure 3C, *p* = 0.82). Thus, irrespective of the target T cell line used, administration of Vβ antibody in conjunction with CD64 IR T cells resulted in statistically significant decreased tumor growth when compared to untransduced T cells. Additionally, use of either Vβ antibody with untransduced T cells did not significantly inhibit tumor growth compared to the tumor only control group. Therefore, Vβ-family-specific antibody prevents malignant T cell tumor outgrowth in vivo only when used in conjunction with CD64 IR T cells.

### 3.5. A Subset of Primary PTCL Samples Show αβ TCR Expression with Varied Vβ Family Usage

Finally, in considering the feasibility of targeting TCR Vβ families in the clinic, we next sought to estimate the percent of T cell lymphoma cases that express the α/β TCR and identify the families used. We identified 15 T cell lymphoma tissue specimens in the University of Pennsylvania pathology archives that showed a clonal T cell proliferation by TCR Vβ sequencing. Of these, eight patient specimens showed expression of αβ T cell receptor using the βF1 immunohistochemistry, and the malignant clone was shown to have a productive TCRβ gene rearrangement (Figure 2D). Although limited by small sample size, we confirmed that a subset of T cell lymphomas express αβ TCR in the malignant cells, and that the Vβ family usage is varied, with five families represented by just eight cases (Figure 2E).

## 4. Discussion

That the β chain of human T cell receptors (TCRs) can be grouped into >20 different functional Vβ families has been established for over 30 years, but this knowledge has not been used for therapeutic purposes [22]. In this chapter, we have shown proof of concept that, using a flexible platform of CD64 immune receptor (IR) modified T cells in combination with Vβ-specific monoclonal antibodies, it is possible to differentially target T cells based on their TCR Vβ family. We believe that this may inform the development of novel immunotherapies for the treatment of PTCLs and other T cell malignancies.

There has been recent interest in developing cellular immunotherapies to target malignant T cells, but the selected targets are not ideal. Chimeric antigen receptor (CAR)-modified T cells targeting a variety of T cell targets such as CD5, CD7, and CD3 do result in target T cell cytolysis, but T cell “fratricide” also occurs [8,9,10]. While it is possible to address “fratricide” by using gene editing techniques to abrogate expression of the antigenic target on the CAR T cells themselves [9], normal healthy T cells also expressing the antigenic target are inevitably killed, which may lead to pan T cell depletion and significant immune compromise. CAR T cells have been recently developed against CD30 [23], which is expressed by a subset of PTCLs, chiefly ALCL. ALCL has the best prognosis of the PTCLs, and the CD30 directed antibody drug conjugate brentuximab vedotin is already available for this diagnosis. CD30 is a marker of T cell activation and is expressed by eosinophils, leading to the possibility of on target off tumor cytolysis of activated T cells and eosinophils.

To achieve ultimate specificity to the clonal population of malignant T cells, personalized anti-idiotype directed immunotherapies would be required. This approach has been used successfully in the treatment of some B cell lymphomas but would require manufacturing of a different product for each patient [24,25,26]. Pule and colleagues recognized that, since TCRs utilize one of only two β chain constant domains, targeting a specific TCR β constant region (TRBC1) would be more practical than the anti-idiotype approaches and provide improved specificity compared to targeting a pan T cell antigen. Importantly, they demonstrated that targeting the TCR is feasible, and that virus specific T cells use both constant domains, so cytolysis of T cells bearing one constant domain is predicted to preserve some viral immunity [11]. We note, however, that targeting one of two constant domains of the TCR would still result in the cytolysis of approximately half of the T cell population, substantially decreasing the TCR repertoire in the patient. Furthermore, the abundance of target cells is believed to contribute to the severity of cytokine release syndrome, a known and potentially serious complication of CAR T cell therapy [27]. Our approach of targeting each of the TCR Vβ families allows for family-specific killing that would include the malignant clone in a given patient with very limited cytolysis of healthy T cells, likely reducing cytokine-mediated toxicities associated with therapy compared to less targeted approaches. Even the most highly represented Vβ family only encompasses ~15% of the entire normal T cell repertoire, and so cytolysis of one Vβ family in an individual would not be predicted to significantly affect immunity or contribute to toxicity as much as the less specific approaches.

We recognize that developing CAR T cells specific to all 24 functional Vβ families still represents a significant challenge. In this paper, we demonstrated proof of concept that TCR Vβ-specific cytolysis is possible, using our flexible CD64 IR platform. While the platform itself is unsuitable for clinical application, since CD64 would bind to circulating immunoglobulins in the patient, likely greatly diminishing specificity, the concept of targeting Vβ families may inform the development of other efficacious Vβ-family-directed immunotherapies for the treatment of T cell malignancies.

## 5. Conclusions

We have established TCR Vβ families as targets in the treatment of T cell malignancies. Using a flexible platform of CD64 IR-transduced T cells, we demonstrated pre-clinically that Vβ-family-specific cytolysis may be achieved through redirected effector T cell activation and firing. In contrast to most current immunotherapeutic approaches developed for T cell malignancies that target pan T cell antigens or one of two TCRβ constant regions, targeting the clonal proliferation of malignant T cells expressing one of 24 functional Vβ families would be much more specific and better preserve the vast majority of the T cell repertoire. We verified the expression of αβ TCRs by malignant T cells in patient specimens and determined that Vβ family usage between patients is highly varied. These results suggest that a flexible approach where different monoclonal antibodies may be used to achieve malignant T cell cytolysis may be most practical in clinical care. Alternatively, a whole armamentarium of engineered T cells covering all the TCR Vβ families may be used. In conclusion, our proof-of-principle approach may be used to inform the development of other novel immunotherapies for the treatment of T cell malignancies.

## 6. Patents

D.J.P. has patent filings related to CD64 immune receptor constructs.

## Figures and Tables

**Figure 1 vaccines-08-00631-f001:**
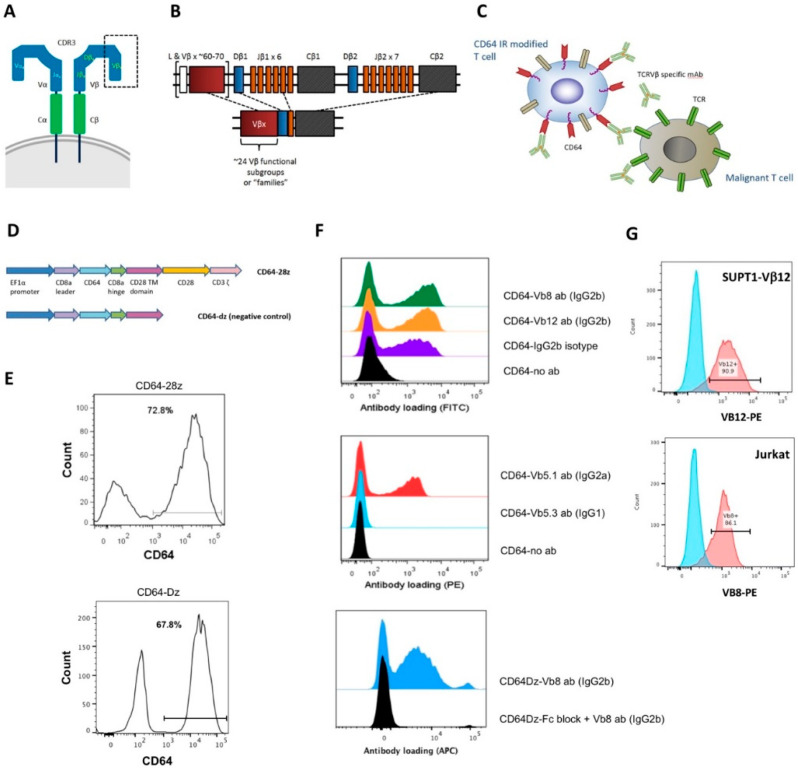
CD64 IR-modified T cells can be directed toward target T cells via TCRVβ specific antibodies. (**A**) Schematic structure of α/β T cell receptor (TCR). (**B**) V(D)J recombination at the TCR β locus. (**C**) Schematic representation of CD64 IR-transduced T cells being directed toward a target malignant T cell via TCRVβ-family-specific mAb. (**D**) Schematic structure of CD64-IR construct. (**E**) Transduction efficiency of primary activated T cells as indicated by staining with CD64 antibody. (**F**) T cells transduced to express CD64-IR were stained with TCR Vβ-directed antibodies of different isotypes and analyzed by flow cytometry. CD64-IR can be loaded with mouse IgG2a and IgG2b mAb, but not IgG1. CD64 dz T cells share same CD64 extracellular domain as CD64 IR and likewise can be loaded with IgG2b mAb. Fc block prevents antibody loading onto CD64 extracellular domain. (**G**) Target cells lines were characterized by staining with TCR Vβ-directed antibodies and analyzed by flow cytometry. The engineered target Sup T1 cell line expresses Vβ12 TCR. Parental SupT1 cells are shown as negative control. Jurkat cells express Vβ8 compared to an isotype control.

**Figure 2 vaccines-08-00631-f002:**
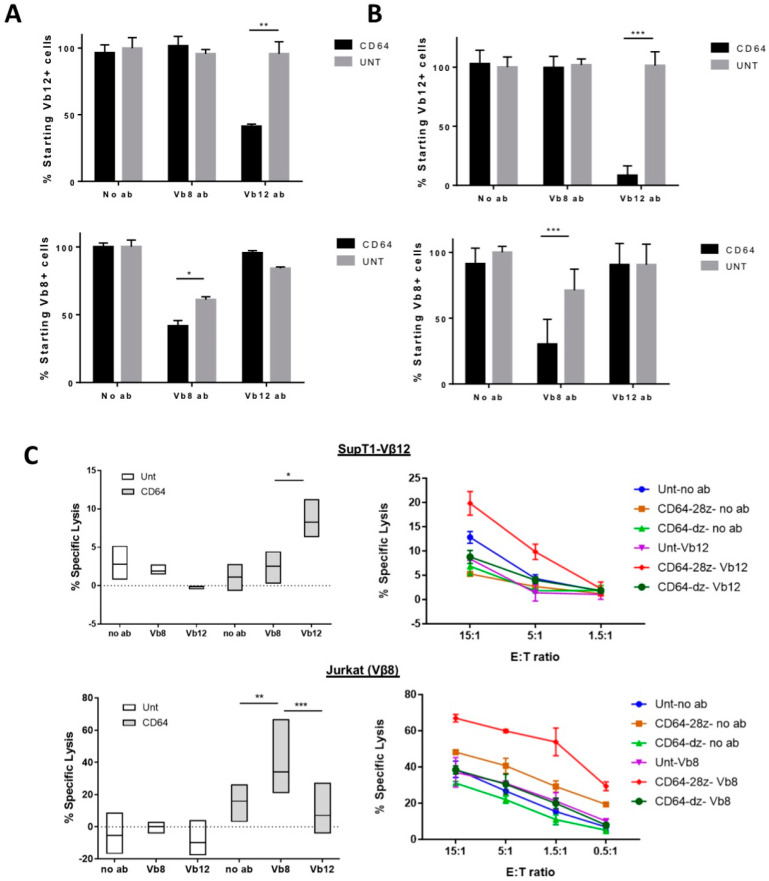
CD64 IR-modified T cells display specific cytolytic function against TCR Vβ families. (**A**,**B**) Autologous lysis of peripheral blood Vβ12 and Vβ8 T cells at 24 h. Vβ-family-specific antibodies were used to “pre-arm” effector T cells (**A**) and “pre-paint” target T cells (**B**). After being co-cultured for 24 h, cells were stained with TCR Vβ-directed antibodies analyzed by flow cytometry to determine the relative change in target cells compared to the starting number. TCR Vβ-specific lysis was observed when CD64-IR cells were redirected against both Vβ8 and Vβ12 cells with both methods. (**C**) Co-culture of T cells and SupT1-Vβ12 and Jurkat T cell line (Vβ8 family) and chromium release assay was performed at 4 h. Left: E:T ratio was 5:1. Data shown for 4 experiments and are normalized to antibody-only control. Right: 4-h chromium release assay performed at different E:T ratios. Technical replicates shown. TCR Vβ-specific lysis was observed when CD64-IR cells were redirected against both Vβ8 and Vβ12 cells. (**D**) Representative H&E staining (top) and T cell antigen receptor βF1 antibody staining (bottom) of paraffin embedded sections of T cell lymphomas. βF1 antibody staining shows cell surface expression of α/β TCR. Intensity of staining ranged from negative to 3+. (**E**) Vβ family expression in eight assessable T cell lymphoma patient specimens shows highly varied TCR Vβ family usage. Vβ family usage is determined by TCRβ NGS, and protein expression is determined by immunohistochemistry (βF1 antibody). Asterisk coding in figures is as follows: * *p* ≤ 0.05; ** *p* ≤ 0.01; *** *p* ≤ 0.001.

**Figure 3 vaccines-08-00631-f003:**
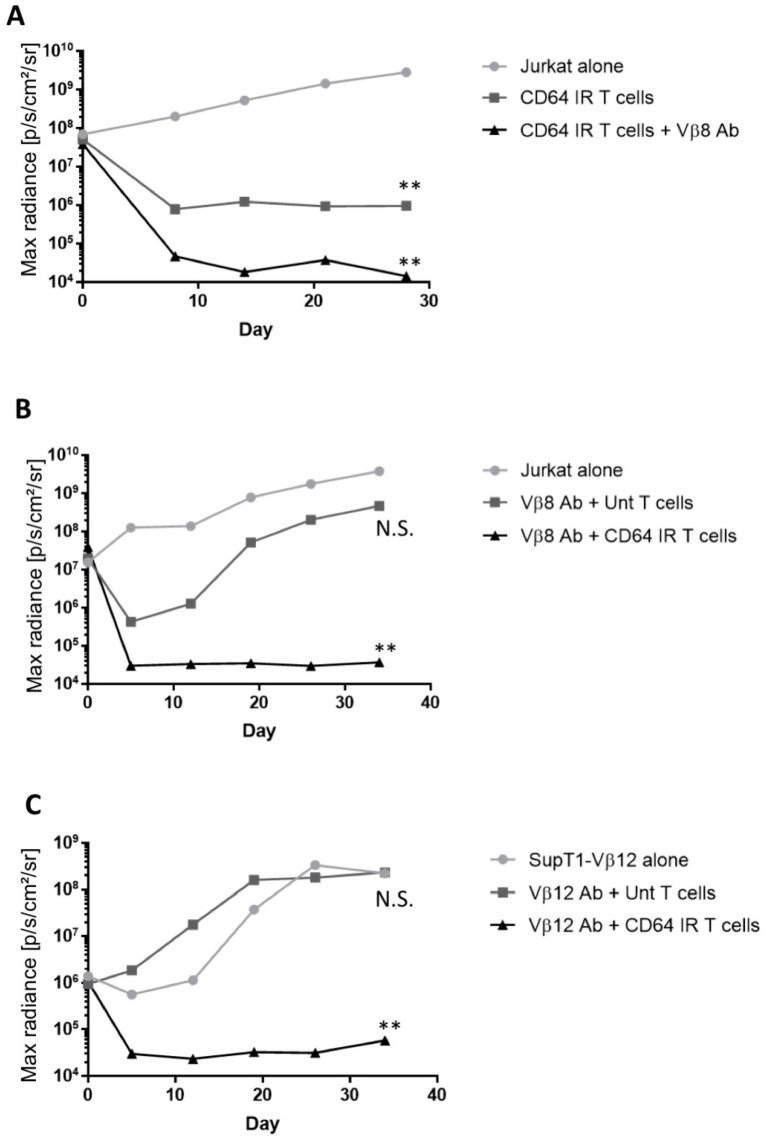
Vβ antibody in conjunction with CD64 IR T cells prevents T cell malignancy outgrowth in vivo. Winn assays are performed by subcutaneous injection of malignant T cell line, Vβ antibody, and T cells as indicated in immunodeficient mice. Median radiance is graphed and significance level indicated by asterisks are in reference to the malignant T cell line only control. (**A**) A pilot Winn assay of CD64 IR T cells and Vβ8 antibody with the Jurkat T cell line. Two mice per group. (**B**) Winn assay of CD64 IR or untransduced T cells and Vβ8 antibody with Jurkat T cell line. Five mice per group. (**C**) Winn assay of CD64 IR or untransduced T cells and Vβ12 antibody with the engineered SupT1-Vβ12 T cell line. Five mice per group. Against the Jurkat cell line, co-administration of CD64 IR T cells slowed the growth of the Jurkat cells compared to mice given tumor only. Addition of Vβ8 antibody was seen to further inhibit tumor growth in this experiment. In all experiments, groups given Vβ antibody in conjunction with CD64 IR T cells resulted in statistically significant decreased tumor growth. Administration of untransduced cells in conjunction with Vβ antibody did not significantly affect tumor growth. Asterisk coding in figures is as follows: * *p* ≤ 0.05; ** *p* ≤ 0.01; *** *p* ≤ 0.001.

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
