# Peer review of "A Novel Approach for the Treatment of T Cell Malignancies: Targeting T Cell Receptor Vβ Families"

_vaccines, 2020, doi:10.3390/vaccines8040631_

Round 1
Reviewer 1 Report
The manuscript (ID): Vaccines-925989, submitted by Wang et al., entitled as “A novel approach for the treatment of T cell malignancies: targeting T cell receptor Vβ families” which demonstrated the concept that TCR Vβ family specific T cell lysis could be achieved by using a novel combination of cell-antibody platform toward the targeting of T cell malignancies without substantial immune compromise, and the manuscript has the outstanding interests in the fields. The manuscript has concisely stated and well written as well. However, I have the following minor concerns.
- Introduction could be improved to give better shape.
- At page-4, line 122, please replace 105 cells/ml with 105 cells/ml and maintain throughout the manuscript.
- It is not clear that how many mice were used to conduct in vivo Please clarify it in the manuscript under the section of MM (In vivo studies).
- As I understood, the authors have tried to see the cytolytic function of CD64 IR T cells against TCR Vβ families and they evaluated only for Vβ12 and/or Vβ8 families. The majority of IFN-γ producing CD4+and CD8+ T-lymphocytes are highly associated with higher frequencies of both T-cells expressing Vβ12 chain though they are also associated with expression of other TCR Vβ chains. So, why authors didn’t try to evaluate other monoclonal antibodies specific against other TCR Vβ families? Clinical trials are necessary to prove this concept fully in the treatment of T cell malignancies– Did authors have plan to conduct clinical trial? Please update it in conclusion.
- In all figures (1, 2 and 3), the legends are too small to see or read out that also cause difficulty to see the statistical analysis as well. Please check and update.
Reviewer 2 Report
Wang et al. showed that a novel approach for the treatment of T cell malignancies by engineering T cell expressing CD64 receptor. This CD64-IR has high affinity to Fc domain of Vbeta antibody and directs this mAb to T cell malignancies. This study is quite interesting and novel. However, few problems need to address.
- The background of CD64 is not enough in Introduction section, which makes reader hard to read.
- The description of Fig. 2D and 2E is quite simple. For TCR Vbeta detecton by NGS, the labeling and explain for Fig. 2E is hard to understand. Moreover, there is no description for specimen collection, IRB and IHC for Fig. 2D and 2E in Method section.
- The figure legend for Fig. 2E needs to clearly state.
- The resolution for Fig. 3 is quite low.
- Fig. 1D showed the negative control CD64-dz. But there is no data in the entire study. Fig. 1E-1G should show the transduction efficiency of CD64-dz and the binding affinity between CD64-dz and Vb antibody.
Round 2
Reviewer 2 Report
I think the sincerity in revisions made us acceptable of the manuscript.